# National Disparities in Antibiotic Prescribing by Race, Ethnicity, Age Group, and Sex in United States Ambulatory Care Visits, 2009 to 2016

**DOI:** 10.3390/antibiotics12010051

**Published:** 2022-12-28

**Authors:** Eric H. Young, Kelsey A. Strey, Grace C. Lee, Travis J. Carlson, Jim M. Koeller, Vidal M. Mendoza, Kelly R. Reveles

**Affiliations:** 1College of Pharmacy, The University of Texas at Austin, Austin, TX 78701, USA; 2Pharmacotherapy Education & Research Center, University of Texas Health San Antonio, San Antonio, TX 78229, USA; 3Fred Wilson School of Pharmacy, High Point University, High Point, NC 27268, USA

**Keywords:** antibiotics, health disparities, epidemiology

## Abstract

While efforts have been made in the United States (US) to optimize antimicrobial use, few studies have explored antibiotic prescribing disparities that may drive future interventions. The objective of this study was to evaluate disparities in antibiotic prescribing among US ambulatory care visits by patient subgroups. This was a retrospective, cross-sectional study utilizing the National Ambulatory Medical Care Survey from 2009 to 2016. Antibiotic use was described as antibiotic visits per 1000 total patient visits. The appropriateness of antibiotic prescribing was determined by ICD-9 or ICD-10 codes assigned during the visit. Subgroup analyses were conducted by patient race, ethnicity, age group, and sex. Over 7.0 billion patient visits were included; 11.3% included an antibiotic prescription. Overall and inappropriate antibiotic prescription rates were highest in Black (122.2 and 78.0 per 1000) and Hispanic patients (138.6 and 79.8 per 1000). Additionally, overall antibiotic prescription rates were highest in patients less than 18 years (169.6 per 1000) and female patients (114.1 per 1000), while inappropriate antibiotic prescription rates were highest in patients 18 to 64 years (66.0 per 1000) and in males (64.8 per 1000). In this nationally representative study, antibiotic prescribing disparities were found by patient race, ethnicity, age group, and sex.

## 1. Introduction

According to the United States (US) Centers for Disease Control and Prevention (CDC), nearly 47 million excess antibiotics are prescribed each year for indications that do not warrant antibiotic use, such as viral infections [1]. Previous studies focusing primarily on the pediatric population or acute respiratory illnesses have found that 30 to 56% of outpatient antibiotic prescriptions are potentially inappropriate or unwarranted [1,2,3,4]. Antibiotic use may increase risk for short- and long-term poor health outcomes, including antibiotic-associated adverse events, antibiotic resistance, and damage to the gut microbiome that puts patients at risk of other complications, such as *Clostridioides difficile* infections [5]. Between 2013 and 2014, antibiotic-associated adverse events accounted for approximately 200,000 emergency department visits in the US, ranking second most common behind anticoagulant-associated adverse events [6].

Given the high rates of outpatient antibiotic prescribing and associated risks to patients and to the global community, it is important to identify prescribing patterns within specific populations who are disproportionately prescribed antibiotics or treated inappropriately for similar conditions. While the impact of heath care disparities has been well established [7,8,9,10], limited studies provide insight into the complex disparities one may encounter when attempting to obtain health care services (access) as well as quality and appropriateness of care [9]. Further adding complexities, certain populations might have a greater risk for certain infectious diseases, experiencing poor antibiotic treatment outcomes, difficulty accessing vaccination programs, and exposure to conditions that promote the emergence of antibiotic resistant infections [11,12].

The CDC and other national organizations have created several initiatives to decrease inappropriate antibiotic use including the National Action Plan for Combating Antibiotic-Resistant Bacteria (CARB) to decrease outpatient prescription dispensing as well as inpatient and outpatient antimicrobial stewardship guidelines [13,14]. While these ongoing efforts are necessary to maximize judicious antibiotic use in the overall US population, further evidence regarding patient populations more likely to be prescribed antibiotics will be another critical step in directing future antimicrobial stewardship efforts. Therefore, the primary aim of this study was to evaluate national disparities in antibiotic prescribing rates in patients of different race, ethnicity, age group, and sex. Additionally, this study aimed to further analyze disparities by presumed appropriateness of antibiotic prescribing. 

## 2. Results

### 2.1. Population Characteristics

Over 7.0 billion visits were included for analysis. In total, 793,415,182 (11.3%) patient visits included a documented antibiotic prescription. Baseline characteristics of patients with and without a documented antibiotic prescription have been described previously [4]. In brief, patients prescribed an antibiotic were younger (median age 41 vs. 51 years) and more often female (58.6% vs. 58.1%), Black (8.7% vs. 7.9%), and Hispanic (12.5% vs. 9.9%). Furthermore, the most common diagnoses where inappropriate antibiotics were prescribed overall were other skin conditions (17.3%), viral upper respiratory infections (13.3%), and bronchitis (11.2%). This trend was reflected in each subgroup as seen in Appendix A, though the most common diagnosis in the age group less than 18 years was in viral upper respiratory infections (26.4%) followed by other skin conditions (20.5%).

### 2.2. Disparities in Antibiotic Prescribing by Race and Ethnicity

Rates of overall and inappropriate antibiotic prescribing by subgroup are provided in Figure 1. Overall and inappropriate antibiotic prescribing rates were highest in Black patients (122.2 and 78.0 per 1000 patient visits, respectively) as well as Hispanic patients (138.5 and 79.8 per 1000 patient visits, respectively). The majority of antibiotics ordered were classified as inappropriate for all races and ethnicities analyzed with only a small percent accounting for antibiotics classified as appropriate for each group (<11%, Table 1). Of patient visits that included an antibiotic, inappropriate prescribing was more common among Black compared to White patients (63.8% vs. 56.2%) and Hispanic compared to non-Hispanic patients (57.5% vs. 56.9%) (Table 1). Additionally, overall prescribing of broad-spectrum antibiotics was highest in White patients and patients of Other race (both 44.8 per 1000 patient visits), followed by Black patients (44.4 per 1000 patient visits). Inappropriate prescribing of broad-spectrum antibiotics was highest in Black patients (28.8 per 1000 patients). Additionally, Hispanic patients had higher overall (57.5 vs. 44.8 per 1000 patient visits) and inappropriate (35.0 vs. 25.8 per 1000 patient visits) prescribing rates of broad-spectrum antibiotics compared to non-Hispanic patients (Appendix A).

### 2.3. Disparities in Antibiotic Prescribing by Age Group

Overall antibiotic prescribing rates were highest for patients less than 18 years old (169.6 per 1000 patient visits) whereas inappropriate antibiotic prescribing rates were highest in patients who were 18 to 64 years old (66.0 per 1000 patient visits, respectively) (Figure 1). Similar trends were seen when analyzing the overall prescribing of broad-spectrum antibiotics. However, rates of inappropriate prescribing of broad-spectrum antibiotic were highest in patients who were 65 years and older (30.3 per 1000 patients) (Appendix A). The overall percentages of antibiotic visits categorized as appropriate were low amongst all age groups (<11.5%, Table 1), whereas the percentage of visits categorized as inappropriate was highest among patients 65 years and older (73.9%), followed by 18 to 64 years (59.9%), and less than 18 years (59.6%) (Table 1).

### 2.4. Disparities in Antibiotic Prescribing by Sex

Female patients had a higher overall rate of antibiotic visits compared to male patients (114.1 vs. 112.3 per 1000 patient visits, respectively), while male patients had a slightly higher rate of inappropriate antibiotic visits compared to females (64.8 vs. 62.4 per 1000 patient visits, respectively) (Figure 1). Of patient visits that included an antibiotic, inappropriate prescribing was more common among male compared to female patients (57.7% vs. 54.7%), though both groups having a small percentage (<11%) of antibiotics classified as appropriate (Table 1). Additionally, male patients had higher overall (46.2 vs. 45.4 per 1000 patient visits) and inappropriate (28.7 vs. 24.4 per 1000 patient visits) prescribing rates of broad-spectrum antibiotics compared to female patients (Appendix A).

## 3. Discussion

In this nationally representative study of ambulatory care office visits in the US between 2009 and 2016, disparities were identified in overall and inappropriate antibiotic prescribing by patient race, ethnicity, age group, and sex. Overall and inappropriate antibiotic prescribing rates were highest in Black and Hispanic patients. Additionally, patients under 18 years of age had the highest overall antibiotic prescription rates while those aged 18 to 64 years had the highest overall inappropriate antibiotic prescribing rates. Lastly, overall antibiotic prescribing was highest in female patients, while inappropriate antibiotic prescribing was slightly higher in male patients. Antibiotic use by subgroup tended to be additive. For example, patients who were classified as female, less than 18 years of age, and Hispanic had a much higher rate of overall antibiotic prescribing compared to others (174.9 per 1000 vs. 112.4 per 1000). Similarly, patients who were classified as male, age 18–64 years, and Hispanic had a much higher rate of inappropriate antibiotic prescribing compared to others (104.1 per 1000 vs. 62.7 per 1000). Identifying disparities in inappropriate antibiotic prescribing practices are crucial to promote and provide optimal patient care and direct antimicrobial stewardship efforts, particularly in the outpatient setting.

When limited to broad-spectrum antibiotics only, Black, Other, and White patients had relatively similar overall antibiotic prescription rates, though Black patients were still found to have the highest prescribing rate of inappropriate broad-spectrum antibiotics. Hispanic patients also had higher overall and inappropriate prescribing rates compared to non-Hispanic patients. Additionally, patients under 18 years of age had the highest overall antibiotic prescription rate while those aged 65 years and older had the highest inappropriate broad-spectrum antibiotic prescribing rates. Lastly, overall and inappropriate antibiotic prescribing was highest in male patients.

Other published studies have evaluated antibiotic prescribing disparities in various populations. In contrast to our findings, some studies have found that Black children are less likely to be prescribed antibiotics or broad-spectrum antibiotics [15,16]. This could be due to the limited pediatric population as well as indication-specific analysis (e.g., otitis media) in these studies. Similarly, in a study of children seen specifically for viral acute respiratory tract infections (ARTIs) in emergency departments, White patients were more likely to receive an antibiotic compared to non-White patients [17]. This contrasting finding could also be due to the more specific patient population studied (e.g., pediatric emergency departments only).

Regarding patient age, one contributing factor as to why children are more likely to receive antibiotics is the high incidence of ARTI development. Research has shown that ARTI diagnoses contribute to more than 70% of visits that result in pediatric antibiotic prescribing [18,19]. Despite this, studies have shown that pediatric patients diagnosed with respiratory infections were less likely to receive inappropriate antibiotics, as pediatricians have been shown to be less likely to inappropriately prescribe antibiotics for ARTIs compared to healthcare providers of other specialties [20]. Patients older than 65 years may have had the highest rates in inappropriate prescribing of broad-spectrum antibiotics due to the often empiric treatment of presumed infection and risk for poor outcomes. One study by Hayward et al. [21] used qualitative interviews of general practitioners in the United Kingdom to better understand antibiotic prescribing practices. It was found that antibiotic prescribing in older adults was more aggressive and included using broad-spectrum antibiotics, longer and earlier duration of treatment, and the use of antibiotics as a diagnostic tool to avoid hospital admissions in patients they perceived to be at a higher risk of death.

Next, our data showed that overall antibiotic prescribing was higher in female patients compared to males, which is consistent with previous literature. In a systematic review by Schröder et al. [22], women were 27% more likely to receive an antibiotic prescription in their lifetime compared to men (PRR 1.27, 95% CI 1.22–1.33). This was further explained in a review by Pinkhasov et al. [23], which showed that women are more likely to consult their health care practitioners more frequently compared to men in a variety of preventive healthcare settings. While our study showed overall antibiotic prescribing rates were higher in female patients, male patients exhibited slightly higher rates of inappropriate antibiotic prescribing. Previous studies have shown that no significant differences exist overall in terms of unnecessary antibiotic prescribing for upper respiratory tract infections between sexes, particularly in conditions such as rhinitis, acute otitis media, acute sinusitis, and acute pharyngotonsillitis [24]. Since data collection for these findings was based on voluntary participation from general practitioners, the results may not be representative of all providers as those interested in appropriate antibiotic use may have been more likely to participate. It is also possible that these differences became more apparent due to the wider scope of diagnoses included in our study.

Socioeconomic, societal, and cultural factors may influence disparities in antibiotic use. Factors may include the patient’s perception of illness, educational level, living conditions, attitudes and expectations, and social pressure [25,26]. A study by McGurn et al. [27] identified socioeconomic status as an important predictor of antibiotic prescribing; children from higher poverty areas were more likely to have received an antibiotic at one week of life compared to children from lower poverty areas even after adjusting for a variety of covariates including age, race, visit type, sex, delivery and birth categories, and chronic medical comorbidities (aOR 1.59, 95% CI 1.02–2.47). Another contributor to antibiotic prescribing rates, particularly in the Hispanic community, could be due to the attitudes and beliefs regarding antibiotic use. In a national study by Francois Watkins et al. (*n* = 7500) [28], Hispanic patients were more likely to believe that antibiotics are helpful in healing from a cold more quickly compared to all other races and ethnicities (48% vs. 25%, respectively). These findings warrant increased education in the role and importance of antibiotic use, particularly in the outpatient setting. Finally, research has shown that there are a myriad of factors that influence antibiotic decision making in the medical setting, including professional influence, such as avoiding challenging colleagues’ prescribing decisions, and communication errors, which could impact patient-provider relationships and judicious antibiotic prescribing [29,30]. In a survey given to over 1500 internal, family, and pediatric medicine physicians in the US, Zetts et al. [31] highlighted that although 91% of participants believed that inappropriate outpatient prescribing was a problem, only 37% believed it was an issue in their own practice setting. Further research on the reasons behind disparate antibiotic prescribing are needed to better inform future public health and clinical guidelines and policies.

The strengths of this study include its large sample size and generalizability to the national U.S. population. This study has potential limitations. For example, the National Ambulatory Medical Care Survey (NAMCS) only provides information on a single office visit; observations from previous visits or longitudinal follow-ups are unavailable. Second, there were only a limited number of diagnoses, medications, and variables collected for each patient visit. As a result, certain pertinent information was not included, such as other additional antibiotic-indicated diagnoses not reported in the database and patient allergies. Third, there is some potential for misclassification bias due to the sole use of diagnosis codes to assign antibiotic appropriateness. Broad-spectrum antibiotics were categorized by antibiotic class; however, this approach may oversimplify spectrum of activity for certain classes (e.g., cephalosporins). Fourth, this study was also not able to account for certain socioeconomic and physician characteristics that could potentially impact antibiotic prescribing. Fifth, an analysis of disparities by geographical region was not performed to limit the scope of the study. Sixth, the NAMCS only accounts for antibiotics that were prescribed in the outpatient setting. Therefore, this study was unable to confirm if antibiotics were filled and taken by patients. Finally, data from the NAMCS excluded any federal, military, and Veterans Affairs related clinics. This exclusion may limit generalizability of this study and underestimate the overall burden of outpatient antibiotic prescribing. Statistically significant findings should also be interpreted with the numeric effect size between groups given the large sample size.

## 4. Materials and Methods

### 4.1. Study Design and Data Source

This was a retrospective, cross-sectional study utilizing the CDC’s NAMCS from 2009 to 2016. The NAMCS collects medical information from a nationally representative sample of patient visits to office-based providers, including those in private practice and free-standing clinics. Each office is assigned an annual one-week reporting period to collect extensive data on a sample of patient visits, including patient demographics, diagnoses codes, medications prescribed, diagnostic procedures performed, and other relevant clinical information. Up to three diagnoses and 8 to 10 medications were collected for outpatient visits between 2009 and 2013, and up to five diagnoses and 30 medications were collected between 2014 and 2016. These data were then used to describe and identify outpatient visits where a patient was prescribed one or more antibiotics.

### 4.2. Patient Population and Study Definitions

All patient visits in the NAMCS from 2009 to 2016 were eligible for inclusion. Antibiotic use was defined as at least one oral antibiotic prescription coded during their office visit using the antibiotic’s Multum code(s) (Appendix A). Topical antibiotic formulations were excluded. To maintain consistency throughout the study years, antibiotics listed in the first eight medication fields of the survey were included. For surveys that included more than eight medications, an additional validation analysis was run to calculate the number of antibiotics documented in columns 9–30. This method was validated by showing that only 4.4% of patients who had a documented antibiotic prescription in the first eight columns also had this documented in columns 9–30. Additionally, none of the patients without antibiotic use documentation in the first eight columns had an antibiotic documented in columns 9–30. Next, antibiotics were classified based on their respective classes (Appendix A). Broad-spectrum antibiotics were defined as cephalosporins, beta-lactam/beta-lactamase inhibitors, and fluoroquinolones. Patient characteristics included age group (less than 18 years, 18 to 64 years, and 65 years and older), sex (male, female), race (Black, White, more than once race, and other), and ethnicity (Hispanic, non-Hispanic), as defined by the NAMCS.

Patient diagnoses were identified using the International Classification of Diseases, 9th Revision, Clinical Modification (ICD-9-CM) codes for survey years 2009 to 2015, and ICD-10 codes for 2016. These codes were limited to the first three fields to maintain consistency throughout all survey years. Additionally, antibiotic use was also identified as appropriate, possibly appropriate, or inappropriate based on the diagnosis coded during the patient visit as previously described by Fleming-Dutra et al. (Appendix A) [1]. In brief, appropriate antibiotic use was defined by a patient visit that included a diagnosis code for a bacterial infection where antibiotic use is almost always indicated (e.g., pneumonia, urinary tract infection). Possibly appropriate antibiotic use was defined by a diagnosis where antibiotics may be indicated (e.g., acne, gastrointestinal infection, pharyngitis, sinusitis, skin infection, otitis media). Finally, inappropriate use included all other visits where a patient visit did not have one of the prior diagnoses and/or had a diagnosis code for a condition where antibiotics are not normally indicated (e.g., asthma/allergy, bronchitis, influenza, viral respiratory infections).

### 4.3. Data and Statistical Analysis

Data analyses were conducted using JMP Pro 16 (SAS Institute, Cary, NC, USA). Data weights were used to extrapolate sample visits to national estimates for all conducted analyses. Baseline characteristics were compared between visits with and without a documented antibiotic prescription using the chi-squared or Wilcoxon rank-sum tests, where appropriate. Antibiotic prescribing rates were calculated as the number of visits that included an antibiotic per 1000 total outpatient visits by patient race, ethnicity, age group, and sex. Prescribing rates in each subpopulation were calculated using population-specific denominators. Appropriateness of prescribing was also described as the rate and percentage of antibiotic visits classified as appropriate, possibly appropriate, and inappropriate in each subgroup. Prescribing rates and appropriateness were compared within groups using the chi-square test.

## 5. Conclusions

In this nationally representative study of ambulatory provider office visits, disparities in overall and inappropriate antibiotic prescribing were found by race, ethnicity, age group, and sex. This study demonstrated that patient characteristics are a potential way for healthcare providers to identify those at risk of inappropriate antibiotic prescribing. While the causes of these disparities are multifactorial, identifying where and why these disparities occur is imperative to further direct antimicrobial stewardship efforts in the outpatient setting. Consideration for the frequency that antibiotics are prescribed and additional socioeconomic and cultural factors will aid in the improvement of these disparities. Furthermore, this study may contribute to protocol development on antibiotic prescribing for patients with characteristics known to put them at risk of inappropriate antibiotic prescribing. Further studies are warranted to continue evaluating antibiotic prescribing trends among various patient populations.

## Figures and Tables

**Figure 1 antibiotics-12-00051-f001:**
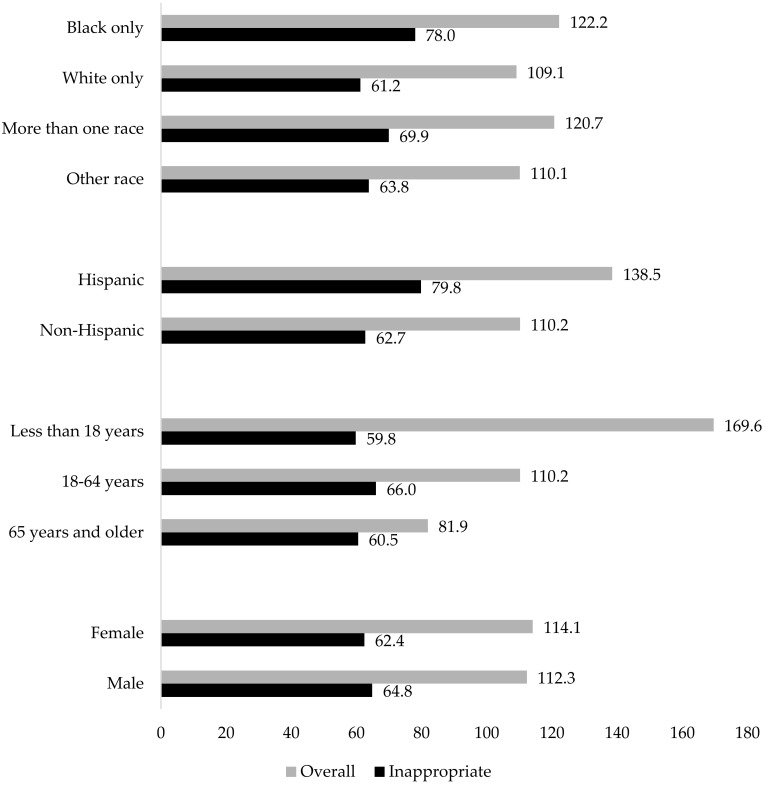
Rates of overall and inappropriate antibiotic prescribing per 1000 total visits by patient subgroup. *p* < 0.0001 for all comparisons of overall and inappropriate antibiotic prescribing rates for each subgroup.

**Table 1 antibiotics-12-00051-t001:** Disparities in inappropriate antibiotic prescribing by patient subgroup.

Characteristic	Inappropriate (*n* = 443,854,893)	Possibly Appropriate (*n* = 283,114,715)	Appropriate (*n* = 66,445,575)
**Race, %**			
Black only	63.8	27.6	8.5
White only	56.2	35.5	8.3
More than one race	57.9	37.5	4.6
Other	58.0	31.6	10.4
**Ethnicity, %**			
Hispanic	57.5	35.0	7.3
Non-Hispanic	56.9	34.5	8.6
**Age group, %**			
Less than 18 years	59.6	35.3	5.1
18 to 64 years	59.9	31.0	9.0
65 years and older	73.9	14.9	11.2
**Sex, %**			
Female	54.7	34.6	10.6
Male	57.7	37.2	5.2

*p* < 0.0001 for all comparisons of percent appropriate prescribing rates for each subgroup.

## Data Availability

Data supporting reported results can be found at: https://www.cdc.gov/nchs/ahcd/datasets_documentation_related.htm (accessed on 2 December 2022).

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
