# Peer review of "National Disparities in Antibiotic Prescribing by Race, Ethnicity, Age Group, and Sex in United States Ambulatory Care Visits, 2009 to 2016"

_antibiotics, 2022, doi:10.3390/antibiotics12010051_

Round 1
Reviewer 1 Report
I have very few suggestions on how to improve the manuscript:
1. Please decipher the abbreviation "NAMES" exactly when it is first mentioned.
2. Add an analysis of the distribution by geographic region in the study's limitations. In chapter 4.3. mentioned that the analysis was carried out by geographical distribution, but the results do not reflect this.
Author Response
Thank you for your review. Please see our responses and manuscript changes attached.

Reviewer 2 Report
There has been a thorough data collection in order to detect disparities in antibiotic prescription across the USA. However, I have some comments which I feel would help render the article more noteworthy:
1. Some more figures could be added in the form of histograms or pie charts to make the reading more inviting given the aumont of data provided.
2. Obviously the author is a native speaker, so the article needn’t any language revision. I did detect, however, what I think is a typo (line 78, should say “as well as Hispanic patients").
3. Most importantly, statistical significance should be calculated, which it can, given the size of the sample, in adittion to other statistical treatment mentioned in section 4.3 Data and statistical analysis, which should appear as a Supplementary table .
4. Also, the conclusions could be further developed with possible measures to counteract the existing breach between subgroups as commented in the discussions, for instance (lines 132-139).
5. Lastly, some of the information in sections like 2.3 Disparities in antibiotic prescribing by age group (lines 98-104 up to "Additionally" essentially provides the same data as the 2nd half of the paragraph, lines 104-110) feel redundant.I think these comments will be helpful to improve the soundness of your manuscript. I would appreciate it if you could ponder them prior to submitting its new version.
Author Response

(The authors gave the same response as above.)

Reviewer 3 Report
Dear Author,
This research is well written and significant to the clinical practice settings; however, minor corrections are required. Please see my comments in the attached file.
Kind regards,

Author Response

(The authors gave the same response as above.)

Reviewer 4 Report
Thank you for allowing me to review this excellent article. As a medical practitioner from a developing nation, I gained valuable insights into antibiotic prescribing practices in a developed nation.
The sample size of seven billion was truly impressive. Also, the analysis was done well and the results were presented splendidly.
The authors have also managed to explain, or at least suggest a plausible explanation for most of their findings. For example, the high incidence of ARTI development children contributed to a higher prescription of antibiotics in this age group. However, why did patients aged 65 years or older have the highest inappropriate broad-spectrum antibiotic prescribing rates?
The authors stated that cephalosporins are broad-spectrum antibiotics. While this is generally true, cephalexin (a 1st generation agent) is more commonly regarded a narrow-spectrum antibiotic.
Author Response

(The authors gave the same response as above.)

Round 2
Reviewer 2 Report
Authors have adequately met the demands.